# OpenReview forum: "Successor Features for Efficient Multi-Subject Controlled Text Generation"
_ICLR.cc/2024/Conference — ICLR 2024 Conference Withdrawn Submission_

### Official Review · Reviewer_bNWc · 2023-10-30

**Soundness:** 3 good
**Presentation:** 1 poor
**Contribution:** 3 good
**Rating:** 6
**Confidence:** 3

**Summary:**

This paper introduces SF-GEN, which is grounded in two primary concepts: successor features (SFs) to decouple the LLM’s dynamics from task-specific rewards, and language model rectification to proportionally adjust the probability of selecting a token based on the
likelihood that the finished text becomes undesired. The result is promising.

**Strengths:**

* The use of successor features is novel for controllable NLG and provides benefits like adding/removing control dimensions efficiently.
* Requires simpler training than methods like discriminator guides or adapter tuning.
* Achieves strong performance - on par or better than various baselines.
* More efficient in memory and computation compared to other methods.

**Weaknesses:**

* The linearity of rewards can limit expressiveness for more complex control objectives.
* Not as performant as state-of-the-art methods like RECT for single dimension control.
* Limited analysis of how it handles multiple simultaneous control dimensions.

**Questions:**

NA

---

> ### Author Response · Authors · 2023-11-17
>
> Thank you for your feedback!
>
> 1. __"The linearity of rewards can limit expressiveness for more complex control objectives."__
> First, we should remark that using our novel reformulation of SF, the reward is only needed at the terminal states and the possible inaccuracy due to linear regression has significantly less impact (as compared to the standard SF formulation). Further, it can be shown theoretically that the nonlinearity of reward can be pushed in the features provided that the features are rich enough, which is largely the case in our framework, as the features are also generated by a pretrained LM. Therefore, while generally the linearity constraint is expected to degrade the quality of value functions, as it is also evident from Table 1 & 2, rectification with SF is on par with the RECT method, which does not use a linear reward.
>
> 2. __"Not as performant as state-of-the-art methods like RECT for single dimension control."__
> The claim of this paper is *not* to achieve better performance than RECT, rather is to enable multi-subject control with almost-zero additional computational overhead as compared to RECT, while achieving comparable performance. This is accomplished through the use of successor features, which effectively separate the dynamics of language from task-specific rewards.
>
> 3. __"Limited analysis of how it handles multiple simultaneous control dimensions."__
> In response to your comment, we would like to direct your attention to Section 5.1 of our paper. In this section, we have specifically focused on analyzing the performance of our method when it comes to the fusion of multiple subjects. This analysis is designed to demonstrate how our approach effectively manages multiple control dimensions in a cohesive manner. The results show that our method is capable of handling complex control scenarios where multiple dimensions are combined.

---

### Official Review · Reviewer_NYoH · 2023-11-01

**Soundness:** 3 good
**Presentation:** 3 good
**Contribution:** 2 fair
**Rating:** 6
**Confidence:** 3

**Summary:**

This paper introduces successor features (SFs) into controllable text generation (CTG) and proposes an efficient decoding framework for multi-subject CTG from the perspective of reinforcement learning (RL). The experimental results on two CTG tasks primarily demonstrate its great performance when compared to other baselines, while maintaining high efficiency. Specifically, in comparison to other methods, the advantage of introducing SFs in this task is evident in its efficiency due to being retraining-free and having lower inference costs (see Strength.2 for more details). In summary, the contributions of this paper are as follows:

1. Building upon previous research that framed LM's generation within the RL framework (Cao et al., 2023), this paper is the first to explore SFs in this research domain and to design a plausible SF-based CTG generation framework.

2. The method introduced in this paper sheds light on the efficiency in the design of CTG tasks, which represents a valuable contribution to GreenNLP.

**Strengths:**

1. **Originality**: This paper presents original research by exploring the application of SFs in multi-subject CTG, utilizing an RL framework and building upon existing theories. This empirical application is the first work in the field of pre-trained LM.
2. **Soundness**: The utilization of SFs offers solutions to address challenges present in previous paradigms:
    - a) The proposed framework leverages SFs to disentangle LM's dynamics from subject rewards, demonstrating flexibility in overcoming the challenges associated with retraining-based methodologies and their associated optimization costs.
    - b) In comparison to other decoding-based methods, the test-time inference cost is reduced, owing to the decreased computational load on tensors.
3. **Significance**: This efficient solution for multi-subject CTG provides valuable insights into how to steer the generation of pre-trained (or large) LMs, with the potential to mitigate bias-related issues without introducing substantial inference latency.

**Weaknesses:**

1. **Performance**: The performance of the proposed method may not be excellent when compared to existing baselines, especially for RECT. To address this concern, the authors could highlight the efficiency of their work in the experiments, in addition to the inference time (discussed in Section 5.2).
2. **Claims**: While the authors mention the application of large LMs in this paper, the main experiments and analysis primarily focus on previous pre-trained LM, specifically GPT-2-large, which lacks instruction-following capabilities. Although GPT4ALL-J is used for prompting experiments, the authors might consider exploring more application scenarios for large LMs. I acknowledge that not all of the CTG paper ought to chase popular large LMs, it is essential to ensure that the claims made regarding large LMs, such as those found in the abstract, are adequately supported through experiments involving large LMs.
3. **Literature review**: Notably, recent parameter-efficient transfer learning methods [1] are used in multi-subject CTG [2]. The authors may consider discussing this paradigm within the paper.
4. **Clarity**: Several typos and clarity issues are present in the paper:
   - The abbreviation "Eq X" is used alongside "Equation X" (page 7) in this paper. It is advisable to standardize the expression.
   - In Section 3: "The state $s_t \in \mathcal{S}$ consists of the prompt and the concatenation of the previously generated tokens." However, it is worth noting that some pre-trained LMs may not take the prompt as their input. The definition of the prompt should be clarified.
   - In Section 3.3, while "SARSA" is a well-known concept, the authors should consider citing relevant literature when mentioning it in this paper for the first time.
   - In Section 3.4, "Laroche et al. (2017))." should be corrected to "Laroche et al. (2017)."


**References**:

[1] Neil Houlsby, Andrei Giurgiu, Stanislaw Jastrzebski, Bruna Morrone, Quentin De Laroussilhe, Andrea Gesmundo, Mona Attariyan, and Sylvain Gelly. 2019. Parameter-efficient transfer learning for NLP. In International Conference on Machine Learning, pages 2790–2799.

[2] Kexin Yang, Dayiheng Liu, Wenqiang Lei, Baosong Yang, Mingfeng Xue, Boxing Chen, and Jun Xie. 2023. Tailor: A Soft-Prompt-Based Approach to Attribute-Based Controlled Text Generation. In Proceedings of the 61st Annual Meeting of the Association for Computational Linguistics (Volume 1: Long Papers), pages 410–427, Toronto, Canada. Association for Computational Linguistics.

**Questions:**

1. Section 3.2: How can Eq 10 be simplified to match Eq 3? Is the focus of this simplification on $r_w(s, a, s')$ in Eq 2?

2. Section 3.3: $\phi$ is parameterized by the output of the final layer of pre-trained LMs. Have the differences between various networks been compared? Is there an exploration of whether simpler networks can achieve similar effectiveness?

---

> ### Author Response · Authors · 2023-11-17
>
> Thank you for your feedback!
>
> ## Weakness
> __"The performance of the proposed method may not be excellent when compared to existing baselines, especially for RECT..."__
>
> As you noted, our method's performance, when compared to RECT, may not appear exceptional at first glance. However, it's important to underline that the primary claim of our paper is not to surpass RECT in terms of performance. Instead, our focus is on enabling multi-subject control with minimal additional computational overhead when compared to RECT, while still maintaining performance that is largely on par.
> To achieve this, we employed successor features in our method, which play a crucial role in our approach. These features effectively disentangle the dynamics of language from task-specific rewards, allowing for more versatile control without significant computational cost. This aspect, we believe, is a substantial contribution to the field, offering a novel approach to multi-subject control in language-based tasks.
> Moreover, as mentioned in Section 5.2, our method demonstrates efficiency in terms of inference time, which is a significant consideration in real-world applications. We propose that this efficiency, coupled with the comparable performance to RECT and the unique advantage of almost-zero additional computational overhead for multi-subject control, underscores the practical value of our work.
> We will make sure to highlight these points more explicitly in the revised manuscript to ensure that the readers fully grasp the significance and novelty of our approach, beyond the traditional performance metrics.
>
> __"Claims: While the authors mention the application of large LMs in this paper, the main experiments and analysis primarily focus on previous pre-trained LM..."__
>
> We appreciate your insights and agree that a broader exploration of different LLMs could further strengthen the claims we made about LLMs. We would like to clarify why we chose GPT4ALL-J for our experiments. The primary reason for this selection is its shared vocabulary with the GPT-2 model, which is a requirement for our approach (the policy model and value function should have the same action space). Evaluation results on common sense reasoning benchmarks show that GPT4ALL-J demonstrated performance comparable to LLaMA. Furthermore, regarding the prompting experiments, we also tested the LLaMA model. Interestingly, we observed that LLaMA's performance in prompting detoxification (with a toxicity score of 60.82) was not as effective as GPT4ALL-J's (which scored 56.13). This comparison underscores the relevance of our findings and shows that the results from GPT4ALL-J can be generalized to other LLMs. In light of your feedback, we plan to extend our research further to include a wider range of LLMs in future studies to provide a more comprehensive understanding of the application scenarios for LLMs.
>
> __"Literature Review & Clarity"__
>
> Thank you for highlighting the missing literature and typos. We will address these issues in the next version of the paper.
>
> ## Questions
> 1. __"Section 3.2: How can Eq 10 be simplified to match Eq 3? Is the focus of this simplification on $r_w(s,a,s')$ in Eq 2?"__
> First, we wish to clarify that Eq 10 represents the combined reward as a weighted sum of subject-level rewards, which aligns directly with the formulation in Eq 3. Therefore, no further simplification between these two equations is necessary.
> Moreover, we would like to point out that Eq 10 is a naive approach by simply taking the average of the rewards. As discussed in our paper, this naive approach is problematic since it is insufficient to satisfy the security condition (i.e., Eq 1) for each individual subject as their corresponding rewards are diluted. To address this, we adopt a different strategy, as detailed in Eq 12, where we use the minimum of the action-value functions for each subject.
>
>
> 2. __"Section 3.3: $\phi$ is parameterized by the output of the final layer of pre-trained LMs. Have the differences between various networks been compared? Is there an exploration of whether simpler networks can achieve similar effectiveness?"__
> In our research, we chose to parameterize $\phi$ using the outputs from the final layer of pre-trained language models (LMs) primarily due to their proven effectiveness in feature extraction for language-related tasks. The representation provided by these LMs is essential for our methodology, as it allows a linear regression model to accurately recover terminal rewards.
> While it is theoretically possible to derive $\phi$ from any feature extractor, we think that non-pretrained models, or simpler networks, may not provide the same level of effectiveness. This is primarily due to their limited ability to capture the intricate semantics of generated sentences. Therefore, we did not explore simpler networks for $\phi$ computation, as pre-trained LMs already established a strong baseline in terms of natural language understanding.

---

> > ### Comment · Reviewer_NYoH · 2023-11-22
> > **Responses (Reviewer NYoH)**
> >
> > I appreciate the authors' comprehensive responses addressing my concerns. I want to provide further clarifications and follow-up comments:
> >
> > > Weakness: Performance Discussion
> >
> > I agree with the authors' opinions about performance. I want to emphasize that my critique regarding the results not being state-of-the-art does not imply a rejection of the paper. It is essential to acknowledge this reality. Actually, as I wrote in my earlier comments under the ``Soundness (a)(b)`` and ``Weaknesses-Performance`` subsections, the proposed method contributes to presenting an efficient computational method. I want to reiterate my appreciation for this aspect highlighted in the previous review.
> >
> > > Claims: LLM scope
> >
> > Thank the authors for providing the justifications for their choice of the testbed. I have no further inquiries regarding this aspect. I am looking forward to seeing border explorations in subsequent work.
> >
> > > Q1: Section 3.2, **How can Eq 10 be simplified to match Eq 3?**
> >
> > My previous question may not be very clear. I understand that Eq 3 associates with  Eq 2. What I want to point out is the following text in the paper:
> >
> > The paper, on Page 4, states: **Thus, we can simplify Eq 10 as ... (Eq 3)**.
> >
> > Since the authors also stated in the response: **"... Therefore, no further simplification between these two equations is necessary. ..."**
> >
> > Is the text on Page 4 a typographical error in the paper, given this clarification?
> >
> > > For other questions, literature review, and clarity, the authors addressed my concerns and I have no further comments.
> >
> > Based on the authors' responses, I would like to raise my overall rating from 5 to 6.

---

### Official Review · Reviewer_th5r · 2023-11-04

**Soundness:** 3 good
**Presentation:** 3 good
**Contribution:** 2 fair
**Rating:** 6
**Confidence:** 2

**Summary:**

The paper introduces SF-Gen to tackle controlled text generation without finetuning a LM’s parameters. SF-Gen is based on two key concepts: (1) language model rectification and (2) successor features (SF). (1) learns a value function to adjust token selection probability during decoding to avoid undesired discourse. (2) disentangles the computation of value functions and tasks, requiring only two models (LLM and SF model) regardless of the end tasks.

Experiments are conducted on two tasks (1) sentiment control and (2) LM detoxification where SF-Gen is compared with baselines that also do not require LM retraining.

**Strengths:**

+ A light-weight solution to controlled text generation. No LM retraining needed and only one additional model is maintained for multiple tasks.

+ Comprehensive experimental design and analysis.

**Weaknesses:**

- Compared with baselines, SF-Gen lags behind some approaches such as DExperts in sentiment control and Rect in both sentiment control and detoxification.

**Questions:**

* In the analysis of combining reward parameters in 5.1, at maximum 3 reward parameters are combined. What if more are added? I imagine as the number of subjects get too large, SFs will have insufficient capacity to model them, or there is no interference at all?

* What’s the main rationale for focusing on GPT-2 XL? Would you expect the observation being different when the base LM is switched to a different one from another family (e.g. Llama) or a different scale?

---

> ### Author Response · Authors · 2023-11-17
>
> Thank you for your feedback!
>
> ## Weakness
> __"Compared with baselines, SF-Gen lags behind some approaches such as DExperts in sentiment control and Rect in both sentiment control and detoxification"__
>
> We would like to underscore that our objective isn't solely focused on outperforming SOTA results on the benchmarks. Rather, the main contribution is to have a dynamic fusion with nearly zero computational overhead as well as comparable performance/accuracy. This is accomplished through the use of successor features, which effectively separate the dynamics of language from task-specific rewards.
>
> ## Questions
> 1. __"In the analysis of combining reward parameters in 5.1, at maximum 3 reward parameters are combined. What if more are added?"__
> SF will have the capacity to model them all since successor features are fully decoupled from rewards. From the SF side, it does not matter if there is only one or many subjects. No problem there. Please see the discussion of Section 3.5 (and equation 12, which is how the values are combined). Mathematically, there is **no problem at all** to add as many reward channels as one requires. Computationally, it is not a problem either, since each reward only adds **one row** to the matrix multiplication, happening after the successor features are acquired (i.e., only one forward pass from the SF network is required, regardless of the number of rewards/subjects).
> However, while there is no problem from the SF viewpoint, one should note that adding **too many subjects** concurrently may defeat the purpose of language generation, as the possibility of cutting too much from the LLM’s generated probability will increase. Mathematically, if infinite diverse subjects are added, then the “augmented LLM” will produce a uniform policy (i.e., not selective). The reason is that under way too many subjects, each token could be a bad choice for at least one of them, hence it cannot be preferred, leaving no preference at all. A good practice here can be to prioritize subjects and only keep maximum $M$ subjects at a time, where $M$ largely depends on the properties of the underlying LLM. We should also note that, in our experiments we did not observe such pathological cases while we tried three subjects concurrently.
>
> 2. __"What’s the main rationale for focusing on GPT-2 XL? Would you expect the observation being different when the base LM is switched to a different one from another family (e.g. Llama) or a different scale?"__
> We use GPT2-large as the policy model for the sentiment control and detoxification experiment. GPT2-XL is used for perplexity evaluation. The focus on the GPT LM family in our paper is primarily due to its widespread popularity in the NLP research community. This popularity makes it easier to compare our results with existing baselines, which predominantly use the GPT as their base model. Additionally, our method requires that the base language model and the action-value function share the same action space, which in this context means they must have the same vocabulary. The GPT family of models, including GPT-2 small, GPT-2 large, GPT-2 XL, and GPT-J share the same tokenizer, fulfilling this requirement. While we have concentrated on the GPT family for these reasons, we believe that our approach would be equally applicable to other LM families, such as Llama.

---

### Official Review · Reviewer_WhQL · 2023-11-07

**Soundness:** 3 good
**Presentation:** 3 good
**Contribution:** 2 fair
**Rating:** 6
**Confidence:** 3

**Summary:**

Controlled text generation has emerged as a significant area of interest, especially when Large Language Models (LLMs) achieve remarkable results across broad applications. However, a potential issue is the typical requirement for retraining LLMs when there is a shift in the control target. To address this, the authors introduce SF-GEN, a method built upon two primary concepts: successor features (SFs) and language model rectification. SF-GEN, following the reinforcement learning (RL) framework for text generation, employs SFs to reduce the complexity of Q-value calculations. Meanwhile, SF-GEN seeks to address challenges associated with the application of SFs to text generation, such as the derivation of the Bellman equation, the interdependency of the value function and task-specific rewards, and the expansive action space. Besides, SF-GEN facilitates the concurrent control of multiple aspects by integrating various reward parameters. Comparative experiments conducted in text generation for sentiment control and detoxification show the superiority of SF-GEN over baselines and most current methods with respect to performance, memory efficiency, and computational speed. Subsequent analysis verifies the advantages of leveraging the decoupling effect of SFs in text generation.

**Strengths:**

1. This work appears to be the first application of SFs, traditionally utilized within RL, to the domain of text generation. RL techniques have demonstrated efficacy in addressing specific challenges in NLP, such as RLHF, and this work is one more example.

2. The adaptation of RL techniques for text generation in this paper is convincingly justified. Each component of the proposed method is introduced by articulating the current challenges and limitations, providing a clear reason for the design.

3. The empirical evaluation showcases the superiority of the proposed method, with experiments across two datasets demonstrating enhanced performance, memory efficiency, and inference speed.

**Weaknesses:**

1. While the paper presents an application of SFs for controlled text generation, the core novelty seems incremental. The principal contribution lies in adapting SFs for multiple subject control within text generation tasks. Despite adjustments to tailor RL techniques to a new domain, the foundational aspects of the proposed SF-GEN method primarily rely on pre-existing approaches.

2. The claimed superiority of the proposed SF-GEN method over competing approaches is not consistently demonstrated across all experimental settings. While potential explanations, such as the linearity constraint, are briefly touched upon, the paper does not offer substantial discussion or experimental evidence to corroborate these hypotheses or to fully account for the discrepancies.

3. The scope of the experimental evaluation appears limited, with the evaluation on two datasets that share similarities. The choice to employ different LLMs for each task raises questions about the comparability of the results. The analysis focuses on the detoxification outcomes, which might present an incomplete picture. A more holistic evaluation, such as the training time in addition to the inference time, would contribute to a deeper understanding (e.g., time efficiency from an algorithmic perspective) of the proposed method.

**Questions:**

1. Could further clarification or empirical evidence be provided regarding the influence of the "linearity constraint" on the comparative results with RECT?

2. Similarly, could additional insights be shared about the "safety conditions" that were factored into the comparative results with DEXPERTS?

3. Regarding the combination of reward parameters, Table 4 does not clearly demonstrate the claim of "without affecting the other". Could the authors expand on this with more details to illustrate this aspect?

---

> ### Author Response · Authors · 2023-11-17
>
> Thank you for your feedback!
> ## Weaknesses
>
> __“While the paper presents an application of SFs for controlled text generation, the core novelty seems incremental...”__
> 1. A key novelty in our approach is the formulation for Successor Features (SF) training, which necessitates accuracy from the reward model **exclusively** at the discourse's end. This design allows the linear regression to capture the actual reward with significant accuracy at the endpoint while disregarding less accurate parts of the sentence. This focused approach in SF training is a substantial departure from existing methods, underscoring our method's novelty. It addresses a challenge in text generation where the reward is only accessible after the complete sentence has been generated.
> 2. Furthermore, we believe that the application of SFs for the text generation domain represents a significant step in this field. The rationale behind our method is to disentangle the language model's dynamics from task-specific rewards. This disentanglement hasn’t been explored before and is crucial for efficiently computing value functions across diverse tasks. For instance, as personalized and customized language models gain more attention, our method presents a notable breakthrough in achieving controlled text generation more efficiently.
>
> __"The scope of the experimental evaluation appears limited"__
> 1. We selected the two datasets for evaluation because they are not only representative but also widely utilized in previous research. This decision was made to ensure that our work aligns with established benchmarks in the field, allowing for a direct and meaningful comparison of our results with existing studies.
> 2. We would like to clarify a misunderstanding regarding the use of different Large Language Models (LLMs) for each task. In our experiments, we consistently employed GPT-2 Large as the base language model across all baselines and our proposed method. This approach was meticulously chosen to ensure the comparability of results across different tasks and methods.
> 3. In Section 5.2, we have included a detailed analysis demonstrating the time efficiency of our approach.
>
> ## Questions
> 1. __"Could further clarification or empirical evidence be provided regarding the influence of the "linearity constraint"..."__
> First, we should remark that using our novel reformulation of SF, the reward is only needed at the terminal states and the possible inaccuracy due to linear regression has significantly less impact (as compared to the standard SF formulation). Further, it can be shown theoretically that the nonlinearity of reward can be pushed in the features provided that the features are rich enough, which is largely the case in our problem, as the features are also generated by a LM. This is accentuated since the base reward in our problems of interest is often quite polar (with a high probability the reward is either very close to one or very close to zero). Hence, the weights in the linear product of $r = \phi^T \cdot w$ can take the form of identifiers, when the feature vector is rich enough (more formally, only some elements of $w$ need to be non-zero to light up the features corresponding to the admission of the subject of interest when there is a non-zero reward, and all components of $w$ can be zero when there is no terminal reward). Therefore, while generally the linearity constraint is expected to degrade the quality of value functions, as it is also evident from Table 1 & 2, rectification with SF is on par with the RECT method, which does not use a linear reward.
> We will add a similar discussion to the paper to clarify and to motivate that SF is highly expected to produce results at the level of RECT, while allowing for multi-subject control.
>
> 2. __"Similarly, could additional insights be shared about the "safety conditions" that were factored into the comparative results with DEXPERTS?"__
> In our paper, when we stated that combining reward parameters does not "affect the other," a more precise expression would be that the impact is "less affected" rather than unaffected. This distinction is crucial and we appreciate your pointing out the need for clarity.
> To elaborate, in Table 4, we presented results where multiple attributes such as "attack" and "threat" are combined and targeted simultaneously. In these cases, we observed a significant reduction in the toxic generations corresponding to these targeted attributes. This indicates that the combination of multiple subjects is effective in specifically reducing toxicity in the targeted categories.
> In the meanwhile, note that attributes not explicitly targeted by the combined parameters, such as "sexually explicit" content, show a different trend. While there is an impact, it is comparatively less affected. This observation is in line with our initial hypothesis that combining reward parameters would primarily influence the targeted attributes, while having a lesser impact on others.

---

> ### Comment · Reviewer_WhQL · 2023-11-20
>
> Thanks for the response. It has helped clarify a few points. My primary concern still stands on the contributions. I intend to keep my original review stance but have updated the subscore to better align with my current understanding of the paper.